# The Applicability of the ESPEN and EASO-Defined Diagnostic Criteria for Sarcopenic Obesity in Japanese Patients after Stroke: Prevalence and Association with Outcomes

**DOI:** 10.3390/nu14194205

**Published:** 2022-10-09

**Authors:** Yoshihiro Yoshimura, Hidetaka Wakabayashi, Fumihiko Nagano, Ayaka Matsumoto, Sayuri Shimazu, Ai Shiraishi, Yoshifumi Kido, Takahiro Bise

**Affiliations:** 1Center for Sarcopenia and Malnutrition Research, Kumamoto Rehabilitation Hospital, Kumamoto 869-1106, Japan; 2Department of Rehabilitation Medicine, Tokyo Women’s Medical University Hospital, Tokyo 162-8666, Japan

**Keywords:** stroke, sarcopenic obesity, rehabilitation, activities of daily living, dysphagia

## Abstract

Sarcopenic obesity is of growing research and clinical interest; however, validated diagnostic criteria are lacking. We therefore aimed to examine the prevalence of sarcopenic obesity as diagnosed by the criteria recently proposed by the European Society for Clinical Nutrition and Metabolism (ESPEN) and the European Association for the Study of Obesity (EASO), and its association with outcomes among patients after stroke. This study was based on a cohort of 760 Japanese patients after stroke admitted to a post-acute rehabilitation hospital. Sarcopenic obesity was diagnosed at admission according to the ESPEN and EASO criteria using reference values specific to Asians. Outcomes included the motor domain of the functional independence measure (FIM-motor) and the food intake level scale (FILS) at discharge. Multivariate linear regression models were used to assess the associations between sarcopenic obesity and outcomes. Among 760 patients (median age, 73 years; 352 women and 408 men), sarcopenic obesity was diagnosed in 34 patients (4.5%; 5.4% of women and 4.1% of men). In multivariate analyses, sarcopenic obesity was independently and negatively associated with FIM-motor (β = −0.048, *p* = 0.031) and FILS at discharge (β = −0.095, *p* = 0.046) in women. In contrast, in men, sarcopenic obesity showed an independent negative association with FIM-motor at discharge (β = −0.117, *p* < 0.001) but no statistically significant association with FILS at discharge (β = −0.004, *p* = 0.323). In conclusion, the prevalence of sarcopenic obesity diagnosed by the ESPEN and EASO-defined criteria was as low as 4.5% among Japanese patients after stroke. Furthermore, sarcopenic obesity was negatively associated with improvements in activities of daily living and dysphagia.

## 1. Introduction

Sarcopenic obesity, characterized by the coexistence of sarcopenia and obesity, has gained increased interest in both research and clinical practice [1]. Sarcopenic obesity, compared with sarcopenia alone or obesity alone, tends to increase the risk of negative health-related outcomes, including falls, comorbidities, physical dependency, frailty, institutionalization, and mortality, in various highly prevalent disease conditions and mortality in the general population, especially in the older population [2,3,4,5]. However, despite the growing interest in sarcopenic obesity in geriatric populations, information on sarcopenic obesity and its potential impact on functional outcomes in geriatric rehabilitation settings is lacking.

Activities of daily living (ADL) and dysphagia are important outcomes of rehabilitation; both are directly related to return-to-home rates for hospitalized patients and quality of life [6,7,8,9]. Sarcopenia, defined as low skeletal muscle mass and function, has been demonstrated to be an independent risk factor for decreased ADL, dysphagia, length of hospital stays, and return-to-home rates among inpatients undergoing rehabilitation [10,11,12,13]. Some studies have shown that sarcopenic obesity is negatively associated with rehabilitation outcomes [5,14]. In contrast, obesity and a high body mass index (BMI) are positively associated with improved ADL in rehabilitation patients, suggesting an obesity paradox [15,16]. Treatment of sarcopenic obesity includes exercise therapy, such as resistance and aerobic exercise, and nutritional therapy, such as energy restriction and high protein intake [17,18]; however, if the diagnosis of sarcopenic obesity is inaccurate, the treatment may not maximize its effectiveness. Furthermore, it is important to differentiate sarcopenic obesity from obesity secondary to endocrine disorders such as hypothyroidism or Cushing’s syndrome. Therefore, it is important to establish a diagnostic and treatment strategy for sarcopenic obesity to improve the quality of rehabilitation medicine and facilitate improvement in ADL and dysphagia in rehabilitation patients.

However, diagnostic criteria for sarcopenic obesity have not been universally established. In previous studies in rehabilitation medicine, the diagnosis of sarcopenia obesity was defined as the presence of both diagnosis of sarcopenia due to decreased skeletal muscle mass and handgrip strength (HGS) and diagnosis of obesity based on increased body fat mass percentage (FM%) using exploratory cut-off values [5,14]. Furthermore, the importance of bioimpedance analysis (BIA) and dual-energy X-ray absorptiometry as body composition assessments, such as skeletal muscle mass and body fat mass, has been reported [19,20,21], but it needs to be verified how to apply them to the diagnosis of sarcopenic obesity in both clinical and research settings. Validated diagnostic criteria for sarcopenia obesity would be useful for patient detection, the comparison of prevalence in different patient groups, and the prediction and assessment of outcomes related to sarcopenic obesity in the general population, as well as in rehabilitation patients. Recently, the European Society for Clinical Nutrition and Metabolism (ESPEN) and the European Association for the Study of Obesity (EASO) proposed a new definition and diagnostic criteria for sarcopenic obesity [22] that need to be validated in rehabilitation settings.

Therefore, we conducted a retrospective cohort study to examine the prevalence of sarcopenic obesity diagnosed using the ESPEN-and EASO-defined criteria and its associations with outcomes such as improvement in ADL and dysphagia in patients undergoing convalescent rehabilitation after stroke.

## 2. Methods

### 2.1. Participants and Setting

We conducted a retrospective cohort study in a post-acute care hospital in Japan to evaluate stroke patients treated at the hospital from January 2016 to December 2020. This hospital is a community-based rehabilitation hospital with 3 convalescent rehabilitation wards, each containing 45 beds (a total of 135 beds). Patients admitted to the wards were divided into three categories according to their disease etiology: stroke, musculoskeletal disorders, or hospital-associated deconditioning. All stroke patients were transferred from the stroke care unit of acute-care hospitals in the local medical cooperation system. 

The exclusion criteria included refusal of consent to participate, missing data, pacemaker implantation, and altered consciousness (indicated by a Japan Coma Scale level of three digits). The observation period was the period of hospitalization (i.e., from the date of admission to the date of discharge).

### 2.2. Convalescent Rehabilitation Program

The convalescent rehabilitation program was tailored to the functional abilities and disabilities of the patient. The program was conducted under the supervision of rehabilitation physicians for a maximum of 3 h per day in accordance with the national medical insurance program. For example, physical therapy includes paralyzed limb facilitation (for leg paralysis), range-of-motion exercises, basic movement training (mainly for the legs and body), walking, resistance (e.g., chair-stand exercises), and ADL trainings [11]. In addition to the individualized structured rehabilitation program, patients underwent a chair-standing exercise, a group-based repetition of the task of sit-to-stand from a chair, in two sessions per day as low-intensity resistance training [23]. Each session lasted 20 min, and the participants were asked to perform a continuous sit-to-stand task up to 120 times at a tempo of about once every 8 s. The frequency and degree of increase in chair-standing exercise varied depending on the ability and endurance of each patient.

Nutritional management, such as aggressive nutritional support, including the provision of high-energy and high-protein meals for malnourished patients, calorie restriction for weight reduction, and the provision of adequate protein for maintaining muscle mass in obese patients, during the hospitalization period was individualized to match the patients’ nutritional and functional statuses under the guidance of registered dietitians and a nutritional support team [24].

Oral management included oral screening, assessment, education, counseling, oral and dysphagia rehabilitation, dental treatment by a dentist, and practice in cooperation with a multidisciplinary team [25,26]. Ward dental hygienists conducted oral and dysphagia rehabilitation, including indirect and direct (oral intake) exercises, at the patient’s bedside [27].

Medication management was performed by multidisciplinary teams including pharmacists [28]. Pharmacotherapy is one of the factors that affect the nutritional status of older people. Polypharmacy and inappropriate medications were corrected and medications that could affect nutritional status were managed throughout the hospital stay [29,30].

### 2.3. Data Collection

Basic patient data were collected upon admission, including age, sex, BMI, stroke type, premorbid ADL using the modified Rankin scale (mRS) score [31], comorbidities using the Charlson comorbidity index (CCI) score [32], days from onset of stroke to admission to the wards [33], information on paralysis using the Brunnstrom recovery stages (BRS) [34], functional independence measure (FIM) scores for physical (FIM-motor) and cognitive (FIM-cognitive) functions [35], nutritional status using the Mini Nutritional Assessment-Short Form (MNA-SF) score [36], dysphagia using the Food Intake Level Scale (FILS) score [37], and days from the onset of stroke [33]. The total number of drugs prescribed at the time of admission was collected from the medical records.

Within 72 h of admission, BIA for fat and skeletal muscle mass, HGS, and the FIM scores for physical (FIM-motor) and cognitive (FIM-cognitive) functions [38] were measured. BIA measurements (InBody S10; InBody Japan Inc., Tokyo, Japan) were performed according to standard procedures to ensure adequate hydration [39]. HGS was measured using a hand dynamometer (Smedley dynamometer; TTM Technologies Inc., Tokyo, Japan) of the nondominant hand (or in case of hemiparesis, of the nonparalyzed hand), with the patients in a standing or seated position (depending on their ability), with their arms straight at their side, and the highest value of the three measurements was recorded. The total rehabilitation therapy received during hospitalization (units per day, 1 unit = 20 min of therapy) was recorded based on the medical charts.

### 2.4. Diagnosis of Sarcopenic Obesity

Patients with sarcopenic obesity were identified according to the definitions and diagnostic criteria of the ESPEN and EASO consensus statements [22]. The evaluation of patients with suspected sarcopenic obesity consisted of two levels: screening and diagnosis. This was followed by a staging evaluation. All analyses including screening, diagnosis, and staging in the diagnosis of sarcopenic obesity were performed by physicians.

#### 2.4.1. Screening

The consensus statement recommends screening for sarcopenic obesity using ethnicity-specific cut-off values for BMI or increased waist circumference in combination with surrogate measures of sarcopenia using clinical symptoms and questionnaires [22]. In the current study, a high BMI and clinical signs were used to screen for obesity and sarcopenia, respectively. The cut-off for high BMI was BMI > 27.5 kg/m^2^ for both sexes, which is the Asian-specific cut-off [40] recommended by the consensus statement [22]. For reference, we also screened for obesity and subsequently diagnosed sarcopenic obesity adopting BMI > 25.0 kg/m^2^, the standard diagnostic cut-off for obesity widely used in clinical practice in Japan [41]. 

Clinical signs for screening for sarcopenia included any of the following: (1) age > 70 years; (2) chronic diseases such as heart failure and kidney disease, or cognitive decline; (3) recent acute illness or nutrition-related events such as recent hospitalization, recent physical inactivity or immobility, recently decreased food intake, or weight loss; or (4) history or complaints of repeated falls, weakness, easy fatigue, or decreased physical activity [22].

#### 2.4.2. Diagnosis

Sarcopenic obesity was diagnosed only in patients who were positive in the above screening. The diagnosis was performed in two stages. First, (1) a decrease in skeletal muscle function was observed. Second, if (1) was found, (2) abnormal body composition was confirmed. 

The consensus statement recommends HGS or chair stand test to assess skeletal muscle function; FM% and low skeletal muscle mass by total skeletal muscle mass adjusted by weight (SMM/W) or appendicular lean mass adjusted by weight (ALM/W), using BIA or dual-energy X-ray absorptiometry, to assess body composition [22]. In this study, a low HGS was used as an indicator of skeletal muscle functional decline. According to the Asia-specific cut-off recommended in the consensus statement [22], the cut-offs for low HGS were set at <28 kg for men and <18 kg for women [42]. In this study, increased FM% and low SMM/W were used as indicators of altered body composition. According to the Asian-specific cut-offs recommended in the consensus statement [22], the cut-offs for high FM% were set at >29.7% for men and >37.2% for women [43], whereas the cut-offs for low SMM/W were <31.5% for men and <22.1% for women [44].

#### 2.4.3. Staging

After the diagnosis of sarcopenic obesity was confirmed, a two-stage staging evaluation was performed. Stage II was defined as the presence of any clinical sign associated with skeletal muscle dysfunction and abnormal body composition, and Stage I was defined as the absence of such signs [22]. Clinical signs included metabolic syndrome, physical dysfunction due to obesity or low muscle mass, cardiovascular, and respiratory diseases.

### 2.5. Outcomes

The primary outcome was the FIM-motor score at discharge, which was evaluated by trained rehabilitation therapists [45,46]. The FIM is divided into a motor domain (FIM-motor) with 13 sub-items and a cognitive domain (FIM-cognitive) with 5 sub-items [38]. Tasks are rated on a seven-point ordinal scale ranging from total assistance to complete independence. The total FIM score ranged from 18 to 126 points, the FIM-motor score ranged from 13 to 91 points, and the FIM-cognitive score ranged from 5 to 35 points. Lower scores indicated dependency. The FIM-motor score at discharge is considered an important outcome of convalescent rehabilitation after stroke [47].

The other outcome was the FILS score at discharge, a validated 10-point observer-rated scale to measure swallowing status [37], which was evaluated by trained nurses.

### 2.6. Sample Size Calculation

The sample size was calculated using data from a previous study [5], the results of which showed that the FIM-motor score of patients admitted to the hospital was normally distributed, with a standard deviation of 26. If the true difference in sample means between those with and without sarcopenic obesity is 17 [48], a sample size of at least 98 participants would be needed in each group to reject the null hypothesis with a power of 0.8 and an alpha error of 0.05, which would support the validity of our results.

### 2.7. Statistical Analysis

Results were reported as means (standard deviations [SDs]) for parametric data, medians and 25th to 75th percentiles (interquartile ranges [IQRs]) for nonparametric data, and numbers (%) for categorical data. The bivariate analysis was based on the presence or absence of sarcopenic obesity and divided into two groups. Between-group comparisons were carried out using a *t*-test, Mann–Whitney U test, or chi-square test depending on the type of variable data.

Multiple linear regression analyses were used to determine whether sarcopenic obesity at admission was independently associated with FIM-motor and FILS scores at discharge. As potential confounders for each outcome, the baseline value (at admission) for each outcome was included as an adjustment factor. In addition, covariates selected to adjust for bias included age, sex, stroke type, stroke history, FIM-motor and -cognitive functions scores, CCI and MNA-SF scores at admission, rehabilitation therapy (units/day), BRS of the lower limb, and length of hospital stay, all of which were considered to be related to the outcomes [49,50,51]. To reduce bias, adjustments for common confounders were performed via a series of multivariate analyses. The multicollinearity was assessed using the variance inflation factor (VIF); a VIF of 1–3 was considered as the absence of multicollinearity. All analyses were performed using SPSS version 21 (IBM Co., Armonk, NY, USA). Statistical significance was set at *p* < 0.05.

### 2.8. Ethics

The study was approved by the institutional review board (IRB) of the hospital (approval ID: 190-220315). Written informed consent was waived by the IRB in view of the retrospective nature of the study and all the procedures being performed were part of the routine care. However, we guaranteed participants’ rights to withdraw from the study using an opt-out procedure. This research was conducted in accordance with the 1964 Helsinki Declaration and its subsequent amendments and with the ethical guidelines of the institutional and national research committee for medical and health research involving human subjects.

## 3. Results

During the study period, 843 stroke patients were newly admitted to the wards. Patients with missing data (*n* = 71), altered consciousness (*n* = 10), or pacemaker implantation (*n* = 2) were excluded. Ultimately, 760 patients were included in the analysis (Figure 1).

Of the 760 patients undergoing convalescent rehabilitation after stroke, 34 (4.5%) met the diagnostic criteria for sarcopenic obesity recommended by the ESPEN and EASO (Figure 2). During the screening, 106 (13.9%) of the 760 patients were in the high BMI category, and all had clinical symptoms related to sarcopenia. In the diagnostic evaluation of patients who screened positive, 40 (40.9%) of 106 patients were in the low HGS category, and 34 of those 40 patients were in the high FM% and low SMM/W categories. Ultimately, 34 of the 760 patients (4.5% total; 5.4% women and 4.1% men) were diagnosed with sarcopenic obesity. At staging, all 34 (4.5%) patients diagnosed with sarcopenic obesity were classified as Stage II. 

As a reference, sarcopenic obesity was found in 60 of 760 patients (7.9% in total: 6.3% in women and 9.3% in men) when a BMI > 25.0 kg/m^2^ was used for obesity screening (Table 1).

The baseline characteristics of the enrolled patients are summarized in Table 2. The median (interquartile range, IQR) age of the patients was 73 (63–81) years, and 46.3% were women. Stroke types included cerebral infarction (*n* = 480, 63.2%), cerebral hemorrhage (*n* = 222, 29.2%), and subarachnoid hemorrhage (*n* = 58, 7.6%). As baseline values for the study outcomes, the median (IQR) FIM-motor score at admission was 49 (21–70), and the median (IQR) FILS score was 8 (7–10). The results of a between-group comparison of patients with and without sarcopenic obesity showed that in women only, patients with sarcopenic obesity had significantly lower FIM-motor (16 (3–22) vs. 54 (19–70), *p* = 0.001) and FIM-cognitive (20 (8–22) vs. 22 (15–28), *p* = 0.025) scores at admission and a longer hospital stay (131 (98–141) vs. 88 (52–145), *p* < 0.001) than those without sarcopenic obesity.

The results of the univariate analysis of the study outcomes between patients with and without sarcopenic obesity are shown in Table 3. Women with sarcopenic obesity had significantly lower FIM-motor scores at discharge than those without (77.1 (40.2–80.4) vs. 83.0 (57.3–88.8), *p* = 0.019). Other outcome measures, such as the FIM-motor score at discharge in men and the FILS score at discharge in men and women, did not significantly differ between groups.

The multivariate linear regression analyses for FIM-motor and FILS scores at discharge, after adjusting for potential confounders including diseases, are shown in Table 4. Both analyses were performed separately for men and women, and there was no multicollinearity between variables. Results showed that sarcopenic obesity was independently and negatively associated with the FIM-motor (β = −0.048, *p* = 0.031) and FILS (β = −0.095, *p* = 0.046) scores at discharge in women. In contrast, in men, sarcopenic obesity showed an independent negative association with the FIM-motor score at discharge (β = −0.117, *p* < 0.001) but no statistically significant association with the FILS score at discharge.

As a reference, Table 5 shows the results of multivariate analyses examining the association between sarcopenic obesity and outcomes when BMI > 25.0 kg/m^2^ instead of BMI > 27.5 kg/m^2^ was used for obesity screening in the diagnosis of sarcopenic obesity. The variables and analytical models used to adjust for confounding factors were the same as those in the series of analyses, and there was no multicollinearity among the variables. Sarcopenic obesity was independently and negatively associated with the FIM-motor (β = −0.073, *p* = 0.024) and FILS scores at discharge (β = −0.096, *p* = 0.042) in women. In contrast, in men, sarcopenic obesity showed an independent negative association with the FIM-motor score at discharge (β = −0.147, *p* = 0.002) but no statistically significant association with the FILS score at discharge. The conclusions were the same as when BMI > 27.5 kg/m^2^ was used for obesity screening, but the association between sarcopenic obesity and each outcome was stronger in both cases when BMI > 25.0 kg/m^2^ was used for obesity screening.

## 4. Discussion

This study aimed to examine the prevalence of sarcopenic obesity diagnosed using the ESPEN and EASO-defined criteria and its associations with outcomes in patients undergoing convalescent rehabilitation after stroke. As a result, three new findings were obtained: (1) the prevalence of sarcopenic obesity diagnosed by the ESPEN and EASO-defined criteria is 4.5%; (2) sarcopenic obesity diagnosed by the ESPEN and EASO-defined criteria is negatively associated with an improvement in ADL; (3) sarcopenic obesity diagnosed by the ESPEN and EASO-defined criteria is negatively associated with an improvement in dysphagia in women.

The prevalence of sarcopenic obesity diagnosed by the ESPEN and EASO-defined criteria was 4.5% (5.4% in women and 4.1% in men). In contrast, when BMI > 25.0 kg/m^2^ was used for obesity screening instead of BMI > 27.5 kg/m^2^, the prevalence of patients diagnosed with sarcopenic obesity was 7.9% (6.3% in women and 9.3% in men). The ESPEN and EASO consensus papers recommend BMI > 27.5 kg/m^2^ as an Asian-specific reference value for obesity screening [40] while acknowledging the lack of widely accepted validity for most reference values for the screening and diagnosis of sarcopenic obesity [22]. In the current study, the diagnosis of sarcopenic obesity using a BMI > 25.0 kg/m^2^ for obesity screening was more strongly associated with outcomes than using a BMI > 27.5 kg/m^2^ for obesity screening. Indeed, even with the same BMI, Asians tend to have more body fat than non-Asians; therefore, the World Health Organization recommends a cut-off of 25 kg/m^2^ for obesity in Asian populations [52], which is lower than the general population in the world. Screening should aim at case finding with maximized sensitivity and the highest possible number of at-risk individuals. Furthermore, cut-off values should be validated as predictors of specific outcomes. Therefore, considering the low prevalence and degree of association with outcomes of sarcopenic obesity, we believe that a BMI > 25.0 kg/m^2^ should be used for obesity screening in this setting. However, future studies are needed to validate the cut-off values in a broad setting in obesity screening for the screening and diagnosis of sarcopenic obesity. 

Sarcopenic obesity diagnosed using the ESPEN and EASO criteria was negatively associated with improvement in ADL. The validity of the ESPEN and EASO criteria in diagnosing sarcopenic obesity in this setting has been suggested. Our data suggest that patients with sarcopenic obesity detected on hospital admission are at risk of poor recovery of ADL, regardless of age, sex, stroke type, comorbidities, nutritional status, and other possible confounders, although the degree of association was weak, with a β of −0.048 in women and −0.117 in men, which is consistent with the results of several available studies in this setting [5,14]. Previous reports have indicated that sarcopenia alone is associated with poor rehabilitation outcomes [11,12]; however, sarcopenic obesity has not been studied extensively in this setting. Obese patients with stroke show higher functional recovery in convalescent rehabilitation wards [16]. This is considered typical of the so-called obesity paradox, and thus, the present findings represent a new perspective that underscores the perception of the negative impact of sarcopenic obesity on rehabilitation outcomes.

Sarcopenic obesity diagnosed using the ESPEN and EASO criteria is negatively associated with an improvement in dysphagia in women. To the best of our knowledge, this is the first study to demonstrate a negative association between sarcopenic obesity and dysphagia recovery in hospitalized adults undergoing convalescent rehabilitation. In recent years, the accumulating evidence has revealed that sarcopenia alone is closely associated with dysphagia; this is called “sarcopenic dysphagia” [53,54,55]. Indeed, the available evidence in the rehabilitation setting indicates that sarcopenia alone is negatively associated with dysphagia and its improvement [10,12,56]. Given the findings of a bidirectional, pathogenic interaction between body fat accumulation and reduced skeletal muscle mass and function, and the finding that the negative clinical interaction between obesity and sarcopenia results in a synergistically higher risk of metabolic disease and functional impairment compared with the cumulative risk from each separate condition [57,58], sarcopenic obesity could have a stronger negative association with dysphagia and its improvement than sarcopenia alone. However, future high-quality studies are needed to examine the association between sarcopenic obesity and dysphagia in a wide range of clinical settings, including sex differences.

This study has several limitations. First, it was conducted at a single community-based rehabilitation hospital in Japan, which may limit the generalizability of the results. Further multicenter studies are needed to verify whether similar results can be obtained in diverse populations. Second, owing to the retrospective study design, we were unable to obtain detailed information on whether the quality and quantity of rehabilitation and nutritional therapy provided during hospitalization affected the results. Future high-quality prospective intervention studies that adjust for these confounding factors are needed. Furthermore, the criteria for sarcopenic obesity used in this study are central to the criteria for frailty discussed in a large body of the literature [59,60,61]. Among them, the validity of estimating body composition using anthropometric measures such as abdominal circumference, calf circumference, and hip circumference has been widely reported [62,63]. Future development of more accurate criteria for the diagnosis of sarcopenic obesity using these indices, which can be easily measured in clinical settings, is expected.

## 5. Conclusions

The prevalence of sarcopenic obesity diagnosed by the ESPEN and EASO-defined criteria was as low as 4.5% (5.4% in women and 4.1% in men) among patients undergoing convalescent rehabilitation after stroke in Japan. Furthermore, sarcopenic obesity was negatively associated with improvements in ADL and dysphagia in this setting. The validity of the ESPEN and EASO criteria in diagnosing sarcopenic obesity in this setting has been suggested. Therefore, we propose that the screening and diagnostic criteria for sarcopenic obesity should be modified according to race and setting and that sarcopenic obesity should be evaluated early and managed appropriately for patients undergoing rehabilitation.

## Figures and Tables

**Figure 1 nutrients-14-04205-f001:**
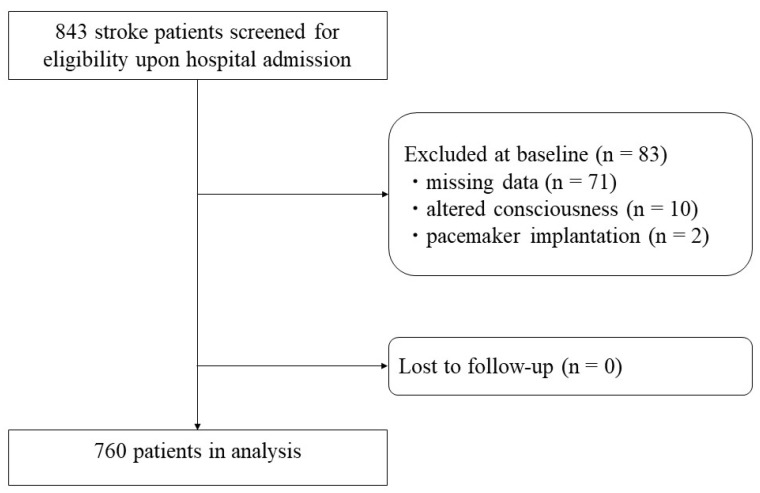
Flowchart of participant screening, inclusion criteria, and follow-up.

**Figure 2 nutrients-14-04205-f002:**
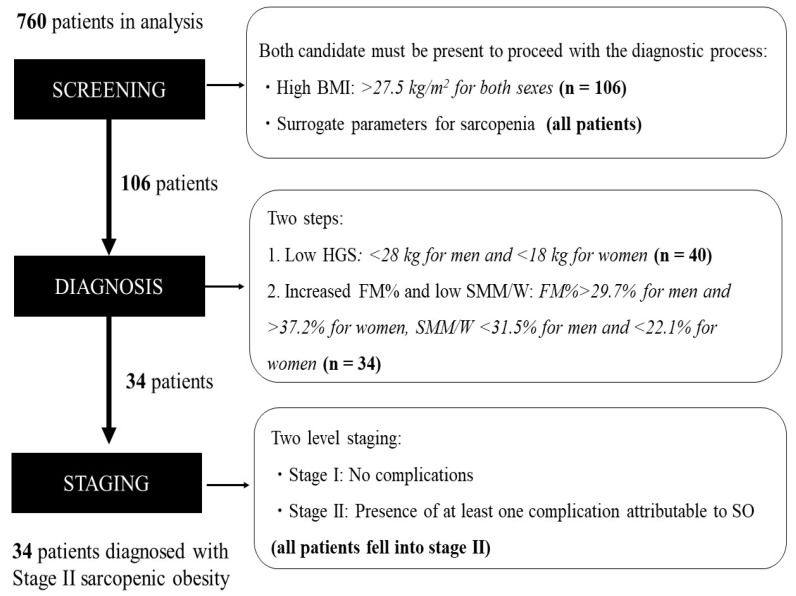
Screening, diagnosis, and staging of sarcopenic obesity according to the ESPEN and EASO-proposed diagnostic criteria. BMI, body mass index; FM, fat mass; HGS, handgrip strength; SMM/W, skeletal muscle mass divided by body weight; SO, sarcopenic obesity.

**Table 1 nutrients-14-04205-t001:** Baseline patient characteristics and univariate analyses for variables between patients with and without sarcopenic obesity defined by ESPEN and EASO criteria 2022 in females and males using BMI > 25.0 kg/m^2^ as the obesity screen.

	Total (*n* = 760)	Female (*n* = 352)			Male (*n* = 408)		
		Without Sarcopenic Obesity (*n* = 330)	With Sarcopenic Obesity (*n* = 22)	*p* Value	Without Sarcopenic Obesity (*n* = 370)	With Sarcopenic Obesity (*n* = 38)	*p* Value
Age, *year*	73 (63, 81)	76 (67, 83)	76 (74, 81)	0.640	68 (60, 79)	76 (62, 82)	0.048
Stroke type, *n* (%)							
-Cerebral infarction	480 (63.2)	190 (57.6)	12 (54.5)	0.826	246 (66.5)	32 (84.2)	0.028
-Cerebral hemorrhage	222 (29.2)	100 (30.3)	10 (45.5)	0.156	106 (28.6)	6 (15.8)	0.125
-SAH	58 (7.6)	38 (11.5)	0 (0.0)	0.149	20 (5.4)	0 (0.0)	0.239
Stroke history, *n* (%)	174 (22.9)	84 (25.5)	2 (9.1)	0.121	72 (19.5)	16 (42.1)	0.003
Premorbid mRS, *score*	0 (0, 1)	0 (0, 1)	0 (0, 1)	0.874	0 (0, 1)	1 (0, 2)	0.042
CCI, *score*	3 (1, 4)	3 (1, 4)	3 (1, 4)	0.591	3 (1, 4)	3 (1, 5)	0.231
Days from onset, *day*	13 (10, 22)	13 (10, 24)	17 (12 23)	0.076	14 (11, 21)	14 (12, 22)	0.804
Paralysis							
Right/Left/Both	336 (44.2)/282 (37.1)/32 (4.2)	148 (44.8)/120 (36.4)/10 (3.0)	10 (45.5)/10 (45.5)/0 (0.0)	0.494	158 (42.7)/138 (37.3)/18 (4.9)	20 (52.6)/14 (36.8)/4 (10.5)	0.303
BRS, score UL/HF/LL	5 (3, 6)/5 (3, 6)/5 (3, 6)	5 (3, 6)/5 (3, 6)/5 (4 6)	5 (2 6)/5 (2 6)/5 (2 6)	0.202	5 (3, 6)/5 (3, 6)/5 (4 6)	4 (1, 5)/4 (2, 5)/5 (1, 5)	0.001
FIM, score							
-total	71 (37, 96)	77 (36, 94)	41 (19, 55)	0.001	72 (41, 97)	47 (27, 63)	<0.001
-motor	49 (21, 70)	54 (20, 70)	16 (13, 40)	0.001	52 (25, 71)	31 (13, 42)	<0.001
-cognitive	22 (14, 28)	22 (15, 28)	20 (7 22)	0.009	22 (14, 30)	15 (12, 25)	<0.001
FILS, *score*	8 (7, 10)	8 (7, 10)	7 (2, 9)	<0.001	8 (7, 10)	7 (6, 9)	0.075
MNA-SF, *score*	7 (5, 9)	8 (6, 9)	7 (5, 9)	0.117	8 (6, 9)	7 (5, 9)	0.170
BMI, kg/m^2^	22.5 (20.2, 25.1)	21.4 (19.2, 24.1)	28.4 (27.7, 30.8)	<0.001	22.7 (21.1, 24.5)	26.7 (25.9, 28.0)	<0.001
-High BMI (> 25.0), *n* (%)	200 (26.3)	64 (19.4)	22 (100.0)	<0.001	76 (20.5)	38 (100.0)	<0.001
-High BMI (> 27.5), *n* (%)	106 (13.9)	34 (10.3)	18 (81.8)	<0.001	38 (10.3)	16 (42.1)	<0.001
SMM/W, %	27.9 (24.0, 31.0)	25.1 (22.5, 27.5)	20.1 (18.5, 21.1)	<0.001	30.8 (28.5, 33.4)	26.5 (23.6, 28.5)	<0.001
-Low SMM/W, *n* (%)	344 (45.3)	62 (18.8)	22 (100.0)	<0.001	222 (60.0)	38 (100.0)	<0.001
FM, %	30.6 (23.8, 36.0)	33.2 (27.6, 38.4)	46.9 (45.8, 48.4)	<0.001	25.8 (20.6, 31.3)	36.9 (33.0, 40.7)	<0.001
-High FM, *n* (%)	280 (36.8)	94 (28.5)	22 (100.0)	<0.001	126 (34.1)	38 (100.0)	<0.001
HG, kg	19.2 (12.5, 28.6)	15.3 (9.2, 19.4)	6.5 (1.4, 15.4)	0.002	28.3 (19.6, 34.6)	16.6 (9.2, 22.9)	<0.001
-Low HG, *n* (%)	456 (60.0)	214 (64.8)	22 (100.0)	<0.001	182 (49.2)	38 (100.0)	<0.001
Number of total drugs	5 (3, 7)	4 (3, 6)	5 (4, 7)	0.154	5 (3, 7)	5 (3, 8)	0.058
Length of hospital stay	95 (56, 145)	88 (52, 145)	157 (109, 163)	<0.001	92 (57, 141)	112 (73, 143)	0.051
Rehabilitation ^a^	8.3 (7.8, 8.6)	8.2 (7.6, 8.5)	8.3 (7.7, 8.6)	0.637	8.3 (7.8, 8.5)	8.3 (8.0, 8.4)	0.346

^a^ Rehabilitation therapy (including physical, occupational, and speech and swallowing therapy) performed during hospitalization (1 unit = 20 min). BMI, body mass index; BRS, Brunnstrom Recovery Stage; CCI, Charlson’s comorbidity index; FILS, Food Intake Level Scale; FIM, Functional Independence Measure; FM, fat mass; HF, hand and finger; HG, handgrip strength; LL, lower limb; mRS, modified Rankin scale; SMI, skeletal muscle mass index; SMM, skeletal muscle mass; UL, upper limb. Data are expressed as means (standard deviation) for parametric data, while medians and 25th to 75th percentiles (interquartile range (IQR)) were used to describe nonparametric data, and numbers (%) were used to describe categorical data.

**Table 2 nutrients-14-04205-t002:** Baseline patient characteristics and univariate analyses for variables between patients with and without sarcopenic obesity defined by ESPEN and EASO criteria 2022 in females and males using BMI > 27.5 kg/m^2^ as the obesity screen.

	Total (*n* = 760)	Female (*n* = 352)			Male (*n* = 408)		
		Without Sarcopenic Obesity (*n* = 334)	With Sarcopenic Obesity (*n* = 18)	*p* Value	Without Sarcopenic Obesity (*n* = 392)	With Sarcopenic Obesity (*n* = 16)	*p* Value
Age, *year*	73 (63, 81])	76 (67, 83)	76 (74, 81)	0.644	72 (60, 80)	72 (61, 79)	0.579
Sex (male), *n* (%)	408 (53.7)	-	-	-	-	-	-
Stroke type, *n* (%)							
-Cerebral infarction	480 (63.2)	192 (57.5)	10 (55.6)	0.999	266 (67.9)	12 (75.0)	0.785
-Cerebral hemorrhage	222 (29.2)	102 (30.5)	8 (44.4)	0.295	108 (27.6)	4 (25.0)	0.989
-SAH	58 (7.6)	38 (11.4)	0 (0.0)	0.238	20 (5.1)	0 (0.0)	0.971
Stroke history, *n* (%)	174 (22.9)	84 (25.1)	2 (11.1)	0.261	80 (20.4)	8 (50.0)	0.088
Premorbid mRS, *score*	0 (0, 1)	0 (0, 1)	0 (0, 1)	0.390	0 (0, 1)	0 (0, 1)	0.452
CCI, *score*	3 (1, 4)	3 (1, 4)	3 (1, 4)	0.946	3 (1, 4)	3 (1, 4)	0.758
Days from onset, *day*	13 (10, 22)	13 (10, 23)	17 (12, 23)	0.149	12 (11, 21)	13 (12, 16)	0.729
Paralysis							
Right/Left/Both	336 (44.2)/282 (37.1)/32 (4.2)	150 (44.9)/122 (36.5)/10 (3.0)	8 (44.4)/8 (44.4)/0 (0.0)	0.617	170 (43.4)/146 (37.2)/20 (5.1)	8 (50.0)/6 (37.5)/2 (12.5)	0.617
BRS, score UL/HF/LL	5 (3, 6)/5 (3, 6)/5 (3, 6)	5 (3, 6)/5 (3, 6)/5 (3, 6)	5 (3, 6)/5 (2, 6)/5 (2, 6)	0.279	5 (3, 6)/5 (3, 6)/5 (3, 6)	5 (3, 5)/5 (3, 5)/5 (2, 5)	0.294
FIM, score							
-total	71 (37, 96)	77 (36, 94)	41 (21, 46)	0.001	71 (41, 96)	53 (41, 72)	0.191
-motor	49 (21, 70)	54 (19, 70)	16 (13, 22)	0.001	49 (24, 70)	32 (25, 54)	0.194
-cognitive	22 (14, 28)	22 (15, 28)	20 (8, 22)	0.025	22 (14, 29)	17 (14, 28)	0.142
FILS, *score*	8 (7, 10)	8 (7, 10)	7 (2, 9)	0.131	8 (7, 10)	7 (7, 10)	0.308
MNASF, *score*	7 (5, 9)	7 (6, 9)	6 (4, 9)	0.015	7 (5, 9)	7 (4, 9)	0.329
BMI, kg/m^2^	22.5 (20.2, 25.1)	21.4 (19.2, 24.3)	28.5 (28.4, 31.0)	<0.001	22.8 (21.3, 24.9)	28.2 (27.8, 29.8)	<0.001
-High BMI (> 25.0), *n* (%)	200 (26.3)	68 (20.4)	18 (100.0)	<0.001	98 (25.0)	16 (100.0)	<0.001
-High BMI (> 27.5), *n* (%)	106 (13.9)	34 (10.2)	18 (100.0)	<0.001	38 (9.7)	16 (100.0)	<0.001
SMM/W, %	27.9 (24.0, 31.0)	25.1 (22.5, 27.5)	20.1 (18.9, 21.1)	<0.001	30.6 (27.9, 33.3)	26.6 (24.5, 28.5)	<0.001
-Low SMM/W, *n* (%)	344 (45.3)	66 (19.8)	18 (100.0)	<0.001	244 (62.2)	16 (100.0)	<0.001
FM, %	30.6 (23.8, 36.0)	33.4 (27.6, 38.7)	46.9 (45.5, 48.7)	<0.001	26.3 (21.0, 32.2)	36.4 (33.0, 39.7)	<0.001
-High FM, *n* (%)	280 (36.8)	98 (29.3)	18 (100.0)	<0.001	148 (37.8)	16 (100.0)	<0.001
HG, kg -Low HG, *n* (%)	19.2 (12.5, 28.6) 456 (60.0)	15.3 (9.1, 19.4) 218 (65.3)	6.3 (0.0, 15.5) 18 (100.0)	0.004 0.001	27.1 (17.5, 34.3) 204 (52.0)	22.8 (15.8, 24.8) 16 (100.0)	0.018 <0.001
Sarcopenic obesity, *n* (%)	34 (4.5)	-	-	-	-	-	-
Number of total drugs	5 (3, 7)	5 (3, 6)	6 (4, 7)	0.075	5 (3, 7)	7 (6, 10)	0.708
Length of hospital stay	95 (56, 145)	88 (52, 145)	131 (98, 141)	<0.001	94 (59, 143)	112 (58, 139)	0.849
Rehabilitation ^a^	8.3 (7.8, 8.6)	8.2 (7.7, 8.5)	8.3 (7.6, 8.5)	0.641	8.3 (7.9, 8.5)	8.3 (8.2, 8.4)	0.897

^a^ Rehabilitation therapy (including physical, occupational, and speech and swallowing therapy) performed during hospitalization (1 unit = 20 min). BMI, body mass index; BRS, Brunnstrom Recovery Stage; CCI, Charlson’s Comorbidity Index; FILS, Food Intake Level Scale; FIM, Functional Independence Measure; FM, fat mass; HF, hand and finger; HG, handgrip strength; LL, lower limb; MNA-SF, Mini Nutritional Assessment-Short Form; mRS, modified Rankin Scale; SMI, skeletal muscle mass index; SMM, skeletal muscle mass; UL, upper limb. Data are expressed as means (standard deviation) for parametric data, while medians and 25th to 75th percentiles (interquartile range (IQR)) were used to describe nonparametric data, and numbers (%) were used to describe categorical data.

**Table 3 nutrients-14-04205-t003:** Univariate analyses for outcomes between patients with and without sarcopenic obesity defined by ESPEN and EASO criteria 2022.

	Total (*n* = 760)	Female (*n* = 352)			Male (*n* = 408)		
		Without Sarcopenic Obesity (*n* = 334)	With Sarcopenic Obesity (*n* = 18)	*p* Value	Without Sarcopenic Obesity (*n* = 392)	With Sarcopenic Obesity (*n* = 16)	*p* Value
FIM-motor at discharge	83 (59, 89)	83.0 (57.3, 88.8)	77.1 (40.2, 80.4)	0.019	87.8 (72.2, 90.5)	80.0 (65.5, 88.7)	0.117
FILS at discharge	10 (9, 10)	10 (9, 10)	9 (9, 10)	0.129	10 (9, 10)	10 (9, 10)	0.334

FILS, Food Intake Level Scale; FIM, Functional Independence Measure. Data are expressed as medians and 25th to 75th percentiles (interquartile range (IQR)) for nonparametric data.

**Table 4 nutrients-14-04205-t004:** Multivariate analyses of sarcopenic obesity defined by ESPEN and EASO criteria 2022 using BMI >27.5 kg/m^2^ as obesity screening for FIM-motor and FILS at discharge in females and males.

	Female				Male			
	FIM-Motor at Discharge		FILS at Discharge		FIM-Motor at Discharge		FILS at Discharge	
	β	*p*	β	*p*	β	*p*	β	*p*
Age	−0.094	0.018	−0.079	0.127	−0.157	0.000	−0.085	0.093
Stroke type								
Cerebral infarction	0.184	0.009	0.080	0.365	0.040	0.674	0.024	0.862
Cerebral hemorrhage	0.221	0.003	0.198	0.030	0.067	469	0.013	0.921
Subarachnoid hemorrhage	(reference)	-	(reference)	-	(reference)	-	(reference)	-
Stroke history	−0.074	0.043	−0.074	0.128	−0.034	0.350	0.089	0.088
FIM-motor at admission	0.430	<0.001	0.365	<0.001	0.410	<0.001	−0.130	0.165
FIM-cognitive at admission	0.192	<0.001	−0.049	0.484	0.194	<0.001	0.280	<0.001
Premorbid mRS	−0.165	<0.001	−0.149	0.005	−0.128	0.001	−0.219	<0.001
CCI	0.085	0.025	0.063	0.216	−0.115	0.002	−0.120	0.026
MNASF at admission	−0.012	0.797	−0.094	0.130	0.160	<0.001	0.116	0.045
Rehabilitation	0.110	0.004	0.187	<0.001	0.059	0.067	0.051	0.257
BRS-lower	0.404	<0.001	-	-	0.214	<0.001	-	-
FILS at admission	-	-	0.466	<0.001	-	-	0.507	<0.001
LOS	0.097	0.059	0.079	0.245	0.194	<0.001	0.191	0.004
Sarcopenic obesity	−0.048	0.031	−0.095	0.046	−0.117	<0.001	−0.004	0.323

BRS, Brunnstrom Recovery Stage; CCI, Charlson’s Comorbidity Index; FILS, Food Intake Level Scale; FIM, Functional Independence Measure; LOS, length of hospital stay; MNA-SF, Mini Nutritional Assessment-Short Form; mRS, modified Rankin scale.

**Table 5 nutrients-14-04205-t005:** Multivariate analyses of sarcopenic obesity defined by ESPEN and EASO criteria 2022 using BMI > 25.0 kg/m^2^ as obesity screening for FIM-motor and FILS at discharge in females and males.

	Female				Male			
	FIM-Motor at Discharge		FILS at Discharge		FIM-Motor at Discharge		FILS at Discharge	
	β	*p*	β	*p*	β	*p*	β	*p*
Age	−0.094	0.018	−0.111	0.033	−0.166	0.000	−0.099	0.052
Stroke type								
Cerebral infarction	0.186	0.009	0.183	0.165	0.048	0.624	0.021	0.876
Cerebral hemorrhage	0.224	0.003	0.326	0.060	0.079	412	0.017	0.961
Subarachnoid hemorrhage	(reference)	-	(reference)	-	(reference)	-	(reference)	-
Stroke history	−0.076	0.039	−0.095	0.048	−0.019	0.607	0.086	0.098
FIM-motor at admission	0.434	<0.001	0.242	0.015	0.398	<0.001	−0.161	0.107
FIM-cognitive at admission	0.187	<0.001	−0.034	0.618	0.193	<0.001	0.296	<0.001
Premorbid mRS	−0.167	<0.001	−0.192	<0.001	−0.140	<0.001	−0.220	<0.001
CCI	0.086	0.023	0.067	0.181	−0.117	0.003	−0.123	0.022
MNASF at admission	−0.010	0.828	−0.128	0.038	0.162	<0.001	0.097	0.094
Rehabilitation	0.111	0.004	0.205	<0.0010	0.059	0.073	0.061	0.182
BRS-lower	0.403	<0.001	0.253	0.001	0.206	<0.001	0.094	0.159
FILS at admission	-	-	0.449	<0.001	-	-	0.471	<0.001
LOS	0.098	0.055	0.125	0.066	0.177	<0.001	0.206	0.002
Sarcopenic obesity	−0.073	0.024	−0.096	0.042	−0.147	0.002	−0.050	0.182

BRS, Brunnstrom Recovery Stage; CCI, Charlson’s Comorbidity Index; FILS, Food Intake Level Scale; FIM, Functional Independence Measure; LOS, length of hospital stay; MNA-SF, Mini Nutritional Assessment-Short Form; mRS, modified Rankin Scale.

## Data Availability

The data are not publicly available owing to opt-out restrictions. Data sharing is not applicable.

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
