# Peer review of "The Applicability of the ESPEN and EASO-Defined Diagnostic Criteria for Sarcopenic Obesity in Japanese Patients after Stroke: Prevalence and Association with Outcomes"

_nutrients, 2022, doi:10.3390/nu14194205_

Round 1

Reviewer 1 Report

Dear Editor, 

Dear Authors,

In my opinion, the topic is interesting, and the manuscript is well written; however, some issues should be addressed.

Major revisions:

INTRODUCTION: This section should be largely improved by highlighting the differential diagnosis with other diseases, the importance of the assessment of the DXA, the disabling sequelae of sarcopenic obesity and the potential rehabilitative treatments.

According to this, you should cite the following references:

-       Dietzel R, Reisshauer A, Jahr S, Calafiore D, Armbrecht G. Body composition in lipoedema of the legs using dual-energy X-ray absorptiometry: a case-control study. Br J Dermatol. 2015 Aug;173(2):594-6. doi: 10.1111/bjd.13697. 

-       Hsu KJ, Liao CD, Tsai MW, Chen CN. Effects of Exercise and Nutritional Intervention on Body Composition, Metabolic Health, and Physical Performance in Adults with Sarcopenic Obesity: A Meta-Analysis. Nutrients. 2019 Sep 9;11(9):2163. doi: 10.3390/nu11092163. 

-       de Sire A, Ferrillo M, Lippi L, Agostini F, de Sire R, Ferrara PE, Raguso G, Riso S, Roccuzzo A, Ronconi G, Invernizzi M, Migliario M. Sarcopenic Dysphagia, Malnutrition, and Oral Frailty in Elderly: A Comprehensive Review. Nutrients. 2022 Feb 25;14(5):982. doi: 10.3390/nu14050982.

-       Petroni ML, Caletti MT, Dalle Grave R, Bazzocchi A, Aparisi Gómez MP, Marchesini G. Prevention and Treatment of Sarcopenic Obesity in Women. Nutrients. 2019 Jun 8;11(6):1302. doi: 10.3390/nu11061302. 

-       Hsu KJ, Liao CD, Tsai MW, Chen CN. Effects of Exercise and Nutritional Intervention on Body Composition, Metabolic Health, and Physical Performance in Adults with Sarcopenic Obesity: A Meta-Analysis. Nutrients. 2019 Sep 9;11(9):2163. doi: 10.3390/nu11092163. 

METHODS: The authors should better clarify who performed the analysis, who supervised the physical therapy sessions, who performed the diagnosis of sarcopenic obesity, and who performed staging evaluation (qualification, degree, and especially if blinded).

METHODS: Physical therapy should be better described including the type of exercises performed, volume (number of sets and repetitions), intensity, and progressions.

Minor revisions:

REFERENCES: References should be formatted following the “Instructions for Authors” of the Journal.

Author Response

Dear editors and reviewers Thank you for providing these comments. We appreciate the time and effort you and each of the reviewers have dedicated to providing insightful feedback on ways to strengthen our paper. We have incorporated changes that reflect the detailed suggestions you have graciously provided. The following is a point-by-point response to the questions and comments. Our changes have marked in red in the revised manuscript. Reviewer 1 Dear Authors, In my opinion, the topic is interesting, and the manuscript is well written; however, some issues should be addressed. (Response) Thank you for your positive comment. Your valuable comments helped us to improve the quality of the manuscript. Major revisions: INTRODUCTION: This section should be largely improved by highlighting the differential diagnosis with other diseases, the importance of the assessment of the DXA, the disabling sequelae of sarcopenic obesity and the potential rehabilitative treatments. According to this, you should cite the following references: Dietzel R, Reisshauer A, Jahr S, Calafiore D, Armbrecht G. Body composition in lipoedema of the legs using dual-energy X-ray absorptiometry: a case-control study. Br J Dermatol. 2015 Aug;173(2):594-6. doi: 10.1111/bjd.13697. Hsu KJ, Liao CD, Tsai MW, Chen CN. Effects of Exercise and Nutritional Intervention on Body Composition, Metabolic Health, and Physical Performance in Adults with Sarcopenic Obesity: A Meta-Analysis. Nutrients. 2019 Sep 9;11(9):2163. doi: 10.3390/nu11092163. de Sire A, Ferrillo M, Lippi L, Agostini F, de Sire R, Ferrara PE, Raguso G, Riso S, Roccuzzo A, Ronconi G, Invernizzi M, Migliario M. Sarcopenic Dysphagia, Malnutrition, and Oral Frailty in Elderly: A Comprehensive Review. Nutrients. 2022 Feb 25;14(5):982. doi: 10.3390/nu14050982. Petroni ML, Caletti MT, Dalle Grave R, Bazzocchi A, Aparisi Gómez MP, Marchesini G. Prevention and Treatment of Sarcopenic Obesity in Women. Nutrients. 2019 Jun 8;11(6):1302. doi: 10.3390/nu11061302. Hsu KJ, Liao CD, Tsai MW, Chen CN. Effects of Exercise and Nutritional Intervention on Body Composition, Metabolic Health, and Physical Performance in Adults with Sarcopenic Obesity: A Meta-Analysis. Nutrients. 2019 Sep 9;11(9):2163. doi: 10.3390/nu11092163. (Response) We appreciate your supportive comment. We agree. We have revised the relevant manuscript in the Introduction according to your advice. We have also cited the references you suggested in the text as follows. (Change) (Introduction-revised) “Sarcopenic obesity, characterized by the coexistence of sarcopenia and obesity, has gained increased interest in both research and clinical practice [1]. Sarcopenic obesity, compared to sarcopenia alone or obesity alone, tends to increase the risk of negative health-related outcomes, including falls, comorbidities, physical dependency, frailty, institutionalization, and mortality, in various highly prevalent disease conditions; mortality in the general population, especially in the older population [2–5]. However, despite the growing interest in sarcopenic obesity in geriatric populations, information on sarcopenic obesity and its potential impact on functional outcomes in geriatric rehabilitation settings are lacking. Activities of daily living (ADL) and dysphagia are important outcomes of rehabilitation; both are directly related to return-to-home rates for hospitalized patients and quality of life [6–9]. Sarcopenia, defined as low skeletal muscle mass and function, has been demonstrated to be an independent risk factor for decreased ADL, dysphagia, length of hospital stays and return-to-home rates among inpatients undergoing rehabilitation [10–14]. Some studies have shown that sarcopenic obesity is negatively associated with rehabilitation outcomes [5,15]. In contrast, obesity and a high body mass index (BMI) are positively associated with improved ADL in rehabilitation patients, suggesting an obesity paradox [16,17]. Treatment of sarcopenic obesity includes exercise therapy, such as resistance and aerobic exercise, and nutritional therapy, such as energy restriction and high protein intake [18,19], however, if the diagnosis of sarcopenic obesity is inaccurate, the treatment may not maximize its effectiveness. Furthermore, it is important to differentiate sarcopenia obesity from obesity secondary to endocrine disorders such as hypothyroidism or Cushing's syndrome. Therefore, it is important to establish a diagnostic and treatment strategy for sarcopenic obesity to improve the quality of rehabilitation medicine and facilitate improvement in ADL and dysphagia in rehabilitation patients. However, diagnostic criteria for sarcopenic obesity have not been universally established. In previous studies in rehabilitation medicine, the diagnosis of sarcopenia obesity was defined as the presence of both diagnosis of sarcopenia due to decreased skeletal muscle mass and handgrip strength (HGS) and diagnosis of obesity based on increased body fat mass percentage (FM%) using exploratory cut-off values [5,15]. Furthermore, the importance of bioimpedance analysis (BIA) and dual-energy X-ray absorptiometry as body composition assessments, such as skeletal muscle mass and body fat mass, has been reported [20–22], which needs to be verified how to apply them to the diagnosis of sarcopenic obesity in both clinical and research settings. Validated diagnostic criteria for sarcopenia obesity would be useful for patient detection, comparison of prevalence in different patient groups, and prediction and assessment of outcomes related to sarcopenic obesity in the general population, as well as in rehabilitation patients. Recently, the Euro-pean Society for Clinical Nutrition and Metabolism (ESPEN) and the European Association for the Study of Obesity (EASO) proposed a new definition and diagnostic criteria for sarcopenic obesity [23] that need to be validated in rehabilitation settings. Therefore, we conducted a retrospective cohort study to examine the prevalence of sarcopenic obesity diagnosed using the ESPEN-and EASO-defined criteria and its association with outcomes such as improvement in ADL and dysphagia in patients undergoing convalescent rehabilitation after stroke.” (Reference) “13. de Sire, A.; Ferrillo, M.; Lippi, L.; Agostini, F.; de Sire, R.; Ferrara, P.E.; Raguso, G.; Riso, S.; Roccuzzo, A.; Ronconi, G.; et al. Sarcopenic Dysphagia, Malnutrition, and Oral Frailty in Elderly: A Comprehensive Review. Nutrients 2022, 14, doi:10.3390/NU14050982. 14. de Sire, A.; Ferrillo, M.; Lippi, L.; Agostini, F.; de Sire, R.; Ferrara, P.E.; Raguso, G.; Riso, S.; Roccuzzo, A.; Ronconi, G.; et al. Sarcopenic Dysphagia, Malnutrition, and Oral Frailty in Elderly: A Comprehensive Review. Nutrients 2022, 14, doi:10.3390/NU14050982.18. Hsu, K.J.; Liao, C. de; Tsai, M.W.; Chen, C.N. Effects of Exercise and Nutritional Intervention on Body Composi-tion, Metabolic Health, and Physical Performance in Adults with Sarcopenic Obesity: A Meta-Analysis. Nutrients 2019, 11, doi:10.3390/NU11092163. 19. Petroni, M.L.; Caletti, M.T.; Grave, R.D.; Bazzocchi, A.; Aparisi Gómez, M.P.; Marchesini, G. Prevention and Treatment of Sarcopenic Obesity in Women. Nutrients 2019, 11, doi:10.3390/NU11061302. 21. Dietzel, R.; Reisshauer, A.; Jahr, S.; Calafiore, D.; Armbrecht, G. Body Composition in Lipoedema of the Legs Us-ing Dual-Energy X-Ray Absorptiometry: A Case-Control Study. Br J Dermatol 2015, 173, 594–596, doi:10.1111/BJD.13697. 22. Batsis, J.A.; Barre, L.K.; Mackenzie, T.A.; Pratt, S.I.; Lopez-Jimenez, F.; Bartels, S.J. Variation in the Prevalence of Sarcopenia and Sarcopenic Obesity in Older Adults Associated with Different Research Definitions: Dual-Energy X-Ray Absorptiometry Data from the National Health and Nutrition Examination Survey 1999-2004. J Am Geriatr Soc 2013, 61, 974–980, doi:10.1111/jgs.12260.” METHODS: The authors should better clarify who performed the analysis, who supervised the physical therapy sessions, who performed the diagnosis of sarcopenic obesity, and who performed staging evaluation. (Response) I appreciate your supportive comment. We agree. We have revised the relevant manuscript according to your advice as follows (Change) (2.2. Convalescent rehabilitation program, Methods) “Convalescent rehabilitation program was tailored to the functional abilities and dis-abilities of the patient. The rehabilitation program was conducted under the supervision of rehabilitation physicians for a maximum of 3 hours per day in accordance with the national medical insurance program. For example, physical therapy includes paralyzed limb facilitation (for leg paralysis), range-of-motion exercises, basic movement training (mainly for the legs and body), walking, resistance (i.e., chair-stand exercises), and ADL trainings [11]. In addition to the individualized structured rehabilitation program, patients underwent “chair-standing exercise,” a group-based repetition of the task of sit-to-stand from a chair in two sessions per day, as low-intensity resistance training [24]. Each session lasted 20 min, and the participants were asked to perform a continuous sit-to-stand task up to 120 times at a tempo of about once every 8 s. The frequency and degree of increase in chair-standing exercise varied depending on the ability and endurance of each patient.” (2.4. Diagnosis of sarcopenic obesity, Methods) “Patients with sarcopenic obesity were identified according to the definitions and di-agnostic criteria of the ESPEN and EASO consensus statements [23]. The evaluation of patients with suspected sarcopenic obesity consisted of two levels: screening and diagnosis. This was followed by a staging evaluation. All analyses, including screening, diagnosis, and staging in the diagnosis of sarcopenic obesity were performed by physicians.” METHODS: Physical therapy should be better described including the type of exercises performed, volume (number of sets and repetitions), intensity, and progressions. (Response) I appreciate your supportive comment. We agree. We have revised the relevant manuscript according to your advice as follows (Change) (2.2. Convalescent rehabilitation program, Methods) “Convalescent rehabilitation program was tailored to the functional abilities and dis-abilities of the patient. The rehabilitation program was conducted under the supervision of rehabilitation physicians for a maximum of 3 hours per day in accordance with the national medical insurance program. For example, physical therapy includes paralyzed limb facilitation (for leg paralysis), range-of-motion exercises, basic movement training (mainly for the legs and body), walking, resistance (i.e., chair-stand exercises), and ADL trainings [11]. In addition to the individualized structured rehabilitation program, patients underwent “chair-standing exercise,” a group-based repetition of the task of sit-to-stand from a chair in two sessions per day, as low-intensity resistance training [24]. Each session lasted 20 min, and the participants were asked to perform a continuous sit-to-stand task up to 120 times at a tempo of about once every 8 s. The frequency and degree of increase in chair-standing exercise varied depending on the ability and endurance of each patient.” Minor revisions: REFERENCES: References should be formatted following the “Instructions for Authors” of the Journal. (Response) Thanks for your comment. We agree. We have revised the references in accordance with the “instructions for Authors”.   Reviewer 2 Let me thank the editor for inviting me to review this excellent study, which however is rather far from any expertise I have, which is mostly “tied” to waist circumference. The paper is very well written, even for what I suspect is a native English speaker. (Response) Thank you for your positive comment. Your valuable comments helped us to improve the quality of the manuscript. It seems to me that the criteria used this study for sarcopenic obesity are central to criteria used for frailty that is addressed in a large literature, which the authors might wish to include in the discussion. Similarly, there are anthropometric indices that address sarcopenia and could be mentioned. (1,2) In future studies, it would be of value to measure waist and hip circumference, with transformation to allometric (mutually independent indices that also can provide non-technical estimation of body composition. (3) 1. Cho HW, Chung W, Moon S, Ryu OH, Kim MK, Kang JG. Effect of Sarcopenia and Body Shape on Cardiovascular Disease According to Obesity Phenotypes. Diabetes Metab J. 2020 Jan;44:e38 2. Krakauer NY, Krakauer JC. Association of Body Shape Index (ABSI) with Hand Grip Strength. Int J Environ Res Public Health. 2020 Sep 17;17(18):E6797. doi: 10.3390/ijerph17186797. PMID: 32957738. 3. Krakauer, N.Y.; Krakauer, J.C. Association of X-ray Absorptiometry Body Composition Measurements with Basic Anthropometrics and Mortality Hazard. Int. J. Environ. Res. Public Health 2021, 18, 7927. (Response) Thank you for your supportive comment. We agree with your concern. We have revised the manuscript according to your comments as follows. (Change) (Last paragraph, Discussion) “This study has several limitations. First, it was conducted at a single community-ty-based rehabilitation hospital in Japan, which may limit the generalizability of the re-suits. Further multicenter studies are needed to verify whether similar results can be obtained in diverse populations. Second, owing to the retrospective study design, we were unable to obtain detailed information on whether the quality and quantity of rehabilitation and nutritional therapy provided during hospitalization affected the results. Future high-quality prospective intervention studies that adjust for these confounding factors are needed. Furthermore, the criteria for sarcopenic obesity used in this study are central to the criteria for frailty discussed in a large literature [60–62]. Among them, the validity of estimating body composition using anthropometric measures such as abdominal circumference, calf circumference and hip circumference has been widely reported [63,64]. Future development of more accurate criteria for the diagnosis of sarcopenic obesity using these indices, which can be easily measured in clinical settings, is expected.” Proof note Table 1 – the stroke types in the first column are not aligned right (Response) Thanks for your positive comment. We agree. We have corrected the alignment of the columns in Tables 1 and 2.

Reviewer 2 Report

B”SD

Initial review 30 September 2022:

Applicability of the ESPEN and EASO-defined Diagnostic Cri teria for Sarcopenic Obesity in Japanese Patients after Stroke:  Prevalence and Association with Outcomes

 Yoshihiro Yoshimura 1*, Hidetaka Wakabayashi 2 , Fumihiko Nagano 1 , Ayaka Matsumoto 1 , Sayuri Shimazu 1 , Ai 5 Shiraishi 1 , Yoshifumi Kido 1 and Takahiro Bise

Let me thank the editor for inviting me to review this excellent study, which however is

rather far from any expertise I have, which is mostly “tied” to waist circumference.  The paper is very well written, even for what I suspect is a native English speaker.

It seems to me that the criteria used this study for sarcopenic obesity are central to  criteria used for frailty that is addressed in a large literature,  which the authors might wish to include in the discussion.  Similarly, there are anthropometric indices that address sarcopenia and could be mentioned. (1,2) In future studies, it would be of value to measure waist and hip circumference, with transformation to allometric (mutually independent indices that also can provide non-technical estimation of body composition. (3)

1.    Cho HW, Chung W, Moon S, Ryu OH, Kim MK, Kang JG.   Effect of Sarcopenia and Body Shape on Cardiovascular Disease According to Obesity Phenotypes.   Diabetes Metab J. 2020 Jan;44:e38

2.    Krakauer NY, Krakauer JC. Association of Body Shape Index (ABSI) with Hand Grip Strength. Int J Environ Res Public Health. 2020 Sep 17;17(18):E6797. doi: 10.3390/ijerph17186797. PMID: 32957738.

3.       . Krakauer, N.Y.; Krakauer, J.C. Association of X-ray Absorptiometry Body Composition Measurements with Basic Anthropometrics and Mortality Hazard. Int. J. Environ. Res. Public Health 2021, 18, 7927.

Annotation: This study brings together allometric anthropometrics and body composition.  Remarkably, directly measured body composition parameters such as %fat can be accurately predicted from height, BMI and ABSI, and overall mortality risk is better predicted by anthropometrics than by body composition.  However, both low limb non-fat tissue (skeletal muscle) and high trunk non fat tissue (perhaps a marker of enlarged internal organs) predict mortality.  The allometric methods in this paper allow combination of body composition and anthropometrics to better estimate mortality risk

Proof note

Table 1 – the stroke types in the first column are not aligned right

Author Response

Dear editors and reviewers

Thank you for providing these comments. We appreciate the time and effort you and each of the reviewers have dedicated to providing insightful feedback on ways to strengthen our paper. We have incorporated changes that reflect the detailed suggestions you have graciously provided. The following is a point-by-point response to the questions and comments. Our changes have marked in red in the revised manuscript.

Reviewer 1

Dear Authors,

In my opinion, the topic is interesting, and the manuscript is well written; however, some issues should be addressed.

(Response)

Thank you for your positive comment. Your valuable comments helped us to improve the quality of the manuscript.

Major revisions:

INTRODUCTION: This section should be largely improved by highlighting the differential diagnosis with other diseases, the importance of the assessment of the DXA, the disabling sequelae of sarcopenic obesity and the potential rehabilitative treatments.

According to this, you should cite the following references:

Dietzel R, Reisshauer A, Jahr S, Calafiore D, Armbrecht G. Body composition in lipoedema of the legs using dual-energy X-ray absorptiometry: a case-control study. Br J Dermatol. 2015 Aug;173(2):594-6. doi: 10.1111/bjd.13697.

Hsu KJ, Liao CD, Tsai MW, Chen CN. Effects of Exercise and Nutritional Intervention on Body Composition, Metabolic Health, and Physical Performance in Adults with Sarcopenic Obesity: A Meta-Analysis. Nutrients. 2019 Sep 9;11(9):2163. doi: 10.3390/nu11092163.

de Sire A, Ferrillo M, Lippi L, Agostini F, de Sire R, Ferrara PE, Raguso G, Riso S, Roccuzzo A, Ronconi G, Invernizzi M, Migliario M. Sarcopenic Dysphagia, Malnutrition, and Oral Frailty in Elderly: A Comprehensive Review. Nutrients. 2022 Feb 25;14(5):982. doi: 10.3390/nu14050982.

Petroni ML, Caletti MT, Dalle Grave R, Bazzocchi A, Aparisi Gómez MP, Marchesini G. Prevention and Treatment of Sarcopenic Obesity in Women. Nutrients. 2019 Jun 8;11(6):1302. doi: 10.3390/nu11061302.

Hsu KJ, Liao CD, Tsai MW, Chen CN. Effects of Exercise and Nutritional Intervention on Body Composition, Metabolic Health, and Physical Performance in Adults with Sarcopenic Obesity: A Meta-Analysis. Nutrients. 2019 Sep 9;11(9):2163. doi: 10.3390/nu11092163.

(Response)

We appreciate your supportive comment. We agree. We have revised the relevant manuscript in the Introduction according to your advice. We have also cited the references you suggested in the text as follows.

(Change)

(Introduction-revised)

Sarcopenic obesity, characterized by the coexistence of sarcopenia and obesity, has gained increased interest in both research and clinical practice [1]. Sarcopenic obesity, compared to sarcopenia alone or obesity alone, tends to increase the risk of negative health-related outcomes, including falls, comorbidities, physical dependency, frailty, institutionalization, and mortality, in various highly prevalent disease conditions; mortality in the general population, especially in the older population [2–5]. However, despite the growing interest in sarcopenic obesity in geriatric populations, information on sarcopenic obesity and its potential impact on functional outcomes in geriatric rehabilitation settings are lacking.

Activities of daily living (ADL) and dysphagia are important outcomes of rehabilitation; both are directly related to return-to-home rates for hospitalized patients and quality of life [6–9]. Sarcopenia, defined as low skeletal muscle mass and function, has been demonstrated to be an independent risk factor for decreased ADL, dysphagia, length of hospital stays and return-to-home rates among inpatients undergoing rehabilitation [10–14]. Some studies have shown that sarcopenic obesity is negatively associated with rehabilitation outcomes [5,15]. In contrast, obesity and a high body mass index (BMI) are positively associated with improved ADL in rehabilitation patients, suggesting an obesity paradox [16,17]. Treatment of sarcopenic obesity includes exercise therapy, such as resistance and aerobic exercise, and nutritional therapy, such as energy restriction and high protein intake [18,19], however, if the diagnosis of sarcopenic obesity is inaccurate, the treatment may not maximize its effectiveness. Furthermore, it is important to differentiate sarcopenia obesity from obesity secondary to endocrine disorders such as hypothyroidism or Cushing's syndrome. Therefore, it is important to establish a diagnostic and treatment strategy for sarcopenic obesity to improve the quality of rehabilitation medicine and facilitate improvement in ADL and dysphagia in rehabilitation patients.

However, diagnostic criteria for sarcopenic obesity have not been universally established. In previous studies in rehabilitation medicine, the diagnosis of sarcopenia obesity was defined as the presence of both diagnosis of sarcopenia due to decreased skeletal muscle mass and handgrip strength (HGS) and diagnosis of obesity based on increased body fat mass percentage (FM%) using exploratory cut-off values [5,15]. Furthermore, the importance of bioimpedance analysis (BIA) and dual-energy X-ray absorptiometry as body composition assessments, such as skeletal muscle mass and body fat mass, has been reported [20–22], which needs to be verified how to apply them to the diagnosis of sarcopenic obesity in both clinical and research settings. Validated diagnostic criteria for sarcopenia obesity would be useful for patient detection, comparison of prevalence in different patient groups, and prediction and assessment of outcomes related to sarcopenic obesity in the general population, as well as in rehabilitation patients. Recently, the Euro-pean Society for Clinical Nutrition and Metabolism (ESPEN) and the European Association for the Study of Obesity (EASO) proposed a new definition and diagnostic criteria for sarcopenic obesity [23] that need to be validated in rehabilitation settings.

Therefore, we conducted a retrospective cohort study to examine the prevalence of sarcopenic obesity diagnosed using the ESPEN-and EASO-defined criteria and its association with outcomes such as improvement in ADL and dysphagia in patients undergoing convalescent rehabilitation after stroke.”

(Reference)

13. de Sire, A.; Ferrillo, M.; Lippi, L.; Agostini, F.; de Sire, R.; Ferrara, P.E.; Raguso, G.; Riso, S.; Roccuzzo, A.; Ronconi, G.; et al. Sarcopenic Dysphagia, Malnutrition, and Oral Frailty in Elderly: A Comprehensive Review. Nutrients 2022, 14, doi:10.3390/NU14050982.

  1. de Sire, A.; Ferrillo, M.; Lippi, L.; Agostini, F.; de Sire, R.; Ferrara, P.E.; Raguso, G.; Riso, S.; Roccuzzo, A.; Ronconi, G.; et al. Sarcopenic Dysphagia, Malnutrition, and Oral Frailty in Elderly: A Comprehensive Review. Nutrients 2022, 14, doi:10.3390/NU14050982.18. Hsu, K.J.; Liao, C. de; Tsai, M.W.; Chen, C.N. Effects of Exercise and Nutritional Intervention on Body Composi-tion, Metabolic Health, and Physical Performance in Adults with Sarcopenic Obesity: A Meta-Analysis. Nutrients 2019, 11, doi:10.3390/NU11092163.
  2. Petroni, M.L.; Caletti, M.T.; Grave, R.D.; Bazzocchi, A.; Aparisi Gómez, M.P.; Marchesini, G. Prevention and Treatment of Sarcopenic Obesity in Women. Nutrients 2019, 11, doi:10.3390/NU11061302.
  3. Dietzel, R.; Reisshauer, A.; Jahr, S.; Calafiore, D.; Armbrecht, G. Body Composition in Lipoedema of the Legs Us-ing Dual-Energy X-Ray Absorptiometry: A Case-Control Study. Br J Dermatol 2015, 173, 594–596, doi:10.1111/BJD.13697.
  4. Batsis, J.A.; Barre, L.K.; Mackenzie, T.A.; Pratt, S.I.; Lopez-Jimenez, F.; Bartels, S.J. Variation in the Prevalence of Sarcopenia and Sarcopenic Obesity in Older Adults Associated with Different Research Definitions: Dual-Energy X-Ray Absorptiometry Data from the National Health and Nutrition Examination Survey 1999-2004. J Am Geriatr Soc 2013, 61, 974–980, doi:10.1111/jgs.12260.

METHODS: The authors should better clarify who performed the analysis, who supervised the physical therapy sessions, who performed the diagnosis of sarcopenic obesity, and who performed staging evaluation.

(Response)

I appreciate your supportive comment. We agree. We have revised the relevant manuscript according to your advice as follows

(Change)

(2.2. Convalescent rehabilitation program, Methods)

Convalescent rehabilitation program was tailored to the functional abilities and dis-abilities of the patient. The rehabilitation program was conducted under the supervision of rehabilitation physicians for a maximum of 3 hours per day in accordance with the national medical insurance program. For example, physical therapy includes paralyzed limb facilitation (for leg paralysis), range-of-motion exercises, basic movement training (mainly for the legs and body), walking, resistance (i.e., chair-stand exercises), and ADL trainings [11]. In addition to the individualized structured rehabilitation program, patients underwent “chair-standing exercise,” a group-based repetition of the task of sit-to-stand from a chair in two sessions per day, as low-intensity resistance training [24]. Each session lasted 20 min, and the participants were asked to perform a continuous sit-to-stand task up to 120 times at a tempo of about once every 8 s. The frequency and degree of increase in chair-standing exercise varied depending on the ability and endurance of each patient.”

(2.4. Diagnosis of sarcopenic obesity, Methods)

Patients with sarcopenic obesity were identified according to the definitions and di-agnostic criteria of the ESPEN and EASO consensus statements [23]. The evaluation of patients with suspected sarcopenic obesity consisted of two levels: screening and diagnosis. This was followed by a staging evaluation. All analyses, including screening, diagnosis, and staging in the diagnosis of sarcopenic obesity were performed by physicians.

METHODS: Physical therapy should be better described including the type of exercises performed, volume (number of sets and repetitions), intensity, and progressions.

(Response)

I appreciate your supportive comment. We agree. We have revised the relevant manuscript according to your advice as follows

(Change)

(2.2. Convalescent rehabilitation program, Methods)

Convalescent rehabilitation program was tailored to the functional abilities and dis-abilities of the patient. The rehabilitation program was conducted under the supervision of rehabilitation physicians for a maximum of 3 hours per day in accordance with the national medical insurance program. For example, physical therapy includes paralyzed limb facilitation (for leg paralysis), range-of-motion exercises, basic movement training (mainly for the legs and body), walking, resistance (i.e., chair-stand exercises), and ADL trainings [11]. In addition to the individualized structured rehabilitation program, patients underwent “chair-standing exercise,” a group-based repetition of the task of sit-to-stand from a chair in two sessions per day, as low-intensity resistance training [24]. Each session lasted 20 min, and the participants were asked to perform a continuous sit-to-stand task up to 120 times at a tempo of about once every 8 s. The frequency and degree of increase in chair-standing exercise varied depending on the ability and endurance of each patient.”

Minor revisions:

REFERENCES: References should be formatted following the “Instructions for Authors” of the Journal.

(Response)

Thanks for your comment. We agree. We have revised the references in accordance with the “instructions for Authors”.

Reviewer 2

Let me thank the editor for inviting me to review this excellent study, which however is rather far from any expertise I have, which is mostly “tied” to waist circumference. The paper is very well written, even for what I suspect is a native English speaker.

(Response)

Thank you for your positive comment. Your valuable comments helped us to improve the quality of the manuscript.

It seems to me that the criteria used this study for sarcopenic obesity are central to criteria used for frailty that is addressed in a large literature, which the authors might wish to include in the discussion. Similarly, there are anthropometric indices that address sarcopenia and could be mentioned. (1,2) In future studies, it would be of value to measure waist and hip circumference, with transformation to allometric (mutually independent indices that also can provide non-technical estimation of body composition. (3)

  1. Cho HW, Chung W, Moon S, Ryu OH, Kim MK, Kang JG. Effect of Sarcopenia and Body Shape on Cardiovascular Disease According to Obesity Phenotypes. Diabetes Metab J. 2020 Jan;44:e38
  2. Krakauer NY, Krakauer JC. Association of Body Shape Index (ABSI) with Hand Grip Strength. Int J Environ Res Public Health. 2020 Sep 17;17(18):E6797. doi: 10.3390/ijerph17186797. PMID: 32957738.
  3. Krakauer, N.Y.; Krakauer, J.C. Association of X-ray Absorptiometry Body Composition Measurements with Basic Anthropometrics and Mortality Hazard. Int. J. Environ. Res. Public Health 2021, 18, 7927.

(Response)

Thank you for your supportive comment. We agree with your concern. We have revised the manuscript according to your comments as follows.

(Change)

(Last paragraph, Discussion)

This study has several limitations. First, it was conducted at a single community-ty-based rehabilitation hospital in Japan, which may limit the generalizability of the re-suits. Further multicenter studies are needed to verify whether similar results can be obtained in diverse populations. Second, owing to the retrospective study design, we were unable to obtain detailed information on whether the quality and quantity of rehabilitation and nutritional therapy provided during hospitalization affected the results. Future high-quality prospective intervention studies that adjust for these confounding factors are needed. Furthermore, the criteria for sarcopenic obesity used in this study are central to the criteria for frailty discussed in a large literature [60–62]. Among them, the validity of estimating body composition using anthropometric measures such as abdominal circumference, calf circumference and hip circumference has been widely reported [63,64]. Future development of more accurate criteria for the diagnosis of sarcopenic obesity using these indices, which can be easily measured in clinical settings, is expected.

Proof note

Table 1 – the stroke types in the first column are not aligned right

(Response)

Thanks for your positive comment. We agree. We have corrected the alignment of the columns in Tables 1 and 2.
